# HIV Replication in Humanized IL-3/GM-CSF-Transgenic NOG Mice

**DOI:** 10.3390/pathogens8010033

**Published:** 2019-03-12

**Authors:** Federico Perdomo-Celis, Sandra Medina-Moreno, Harry Davis, Joseph Bryant, Juan C. Zapata

**Affiliations:** 1Institute of Human Virology, School of Medicine, University of Maryland, Baltimore, MD 21201, USA; fcelis@ihv.umaryland.edu or federico.perdomo@usco.edu.co (F.P.-C.); smmoreno@ihv.umaryland.edu (S.M.-M.); hdavis@ihv.umaryland.edu (H.D.); jbryant@ihv.umaryland.edu (J.B.); 2Grupo Inmunovirología, Facultad de Medicina, Universidad de Antioquia, UdeA, Medellín 050010, Colombia

**Keywords:** NOG mice, Humanized mice (huNOG or huNOG-EXL), GM-CSF, IL-3, HIV

## Abstract

The development of mouse models that mimic the kinetics of Human Immunodeficiency Virus (HIV) infection is critical for the understanding of the pathogenesis of disease and for the design of novel therapeutic strategies. Here, we describe the dynamics of HIV infection in humanized NOD/Shi-scid-IL2rγ^null^ (NOG) mice bearing the human genes for interleukin (IL)-3 and granulocyte-macrophage colony-stimulating factor (GM-CSF) (NOG-EXL mice). The kinetics of viral load, as well as the frequencies of T-cells, B-cells, Natural killer cells (NK), monocytes, and dendritic cells in blood and secondary lymphoid organs were evaluated throughout the time of infection. In comparison with a non-transgenic humanized mouse (NSG) strain, lymphoid and myeloid populations were more efficiently engrafted in humanized NOG-EXL mice, both in peripheral blood and lymphoid tissues. In addition, HIV actively replicated in humanized NOG-EXL mice, and infection induced a decrease in the percentage of CD4^+^ T-cells, inversion of the CD4:CD8 ratio, and changes in some cell populations, such as monocytes and dendritic cells, that recapitulated those found in human natural infection. Thus, the humanized IL-3/GM-CSF-transgenic NOG mouse model is suitable for the study of the dynamics of HIV infection and provides a tool for basic and preclinical studies.

## 1. Introduction

An important limitation for the study of Human Immunodeficiency Virus (HIV) infection pathogenesis is the species-specificity of this virus [1]. Although models of Simian Immunodeficiency Virus (SIV) infection in nonhuman primates have largely contributed to a better knowledge of HIV pathophysiology, disadvantages of their use include the high cost and increased gestation period, limitation to a smaller number of animals, host restriction factors that affect viral replication, and ethical concerns, among other considerations [2,3]. Although mice cannot support HIV infection, mouse humanization via transplantation of CD34^+^ hematopoietic stem cells (HSCs) into immunodeficient strains offers the possibility to study different human diseases after reconstitution of cell populations [4]. In addition, immunodeficient mouse models allow the creation of knock-out or transgenic strains for the study of specific host features and their impact on disease dynamics. For instance, transgenic mice expressing the human cytokines, interleukin (IL)-3 and granulocyte-macrophage colony-stimulating factor (GM-CSF), support the heightened engraftment of myeloid cells [5], which could be useful for the study of the role of these subsets in the pathogenesis or resistance to HIV infection.

Here, we describe the kinetics of T-cells, B-cells, natural killer (NK) cells, monocytes, and myeloid and plasmacytoid dendritic cells (mDCs and pDCs) in humanized NOD/Shi-*scid*-IL2rγ^null^ (NOG) mice [6] transgenic for the human IL-3 and GM-CSF (huNOG-EXL mice) [7], in basal conditions and during HIV infection. Our results indicate that this model recapitulates some features of HIV infection, such as the increase in the viral load, inversion of the CD4:CD8 ratio, and changes in some lymphoid and myeloid populations in peripheral blood and lymphoid tissues, and support its usefulness for basic and preclinical studies.

## 2. Results and Discussion

### 2.1. Characterizing Lymphoid and Myeloid Populations in huNOG-EXL Mice

Through flow cytometry and the use of anti-human monoclonal antibodies, here, we evaluated the engraftment of human lymphoid and myeloid cell populations in huNOG-EXL. We first characterized cell populations in huNOG-EXL mice in basal conditions and compared them with huNSG control mice, after 10–14 weeks post-engraftment. Of note, NSG mice bear a genetic background similar to NOG mice [8], but they exhibit full absence of IL2rγ, in contrast to NOG mice, which contain a non-functional truncated IL2rγ. NSG mice are not transgenic for human IL-3 and GM-CSF [9]. NOG-EXL mice are commercially available from Taconic, and NSG mice were used according to our in-house-standardized conditions [10,11]. NSG mice are a reference strain and are widely used for HIV studies [12,13,14].

T-cells, B-cells, NK cells (CD56^bright^, CD56^dim^, and CD56^−^) [15], monocytes (classical, intermediate, and non-classical) [16], and mDCs and pDCs [17,18] were identified according to the gating strategy shown in Figure 1. In the case of T-cells, we also explored CXCR5-expressing cells as an indirect measurement of the development of lymphoid follicles and cell homing to these structures in secondary lymphoid organs (SLO) [19,20]. As shown in Figure 2, compared with huNSG mice, huNOG-EXL mice had higher levels of engraftment, evaluated by the percentage of human CD45^+^ cells in blood (Figure 2A). huNOG-EXL exhibited lower frequencies of circulating T-cells than huNSG mice, at expenses of higher proportions of B-cells (Figure 2B,C). There were no differences in the CD4:CD8 ratios between both mice strains (Figure 2D). Finally, huNOG-EXL had higher frequencies of circulating CXCR5^+^ CD4^+^ (Figure 2E) and CD8^+^ T-cells (Figure 2F). On the other hand, huNOG-EXL and huNSG mice had comparable frequencies of CD45^+^ cells, T-cells, and B-cells in SLO (spleen, axillary [ALN], and mesenteric lymph nodes [MLN]) (Figure 2A–C), but huNOG-EXL exhibited higher proportions of CXCR5^+^ CD4^+^ T-cells (Figure 2E).

NK cells were evaluated from FSC-A^lo^ CD3^−^ cells. Similar to previous human reports [21,22], in huNOG-EXL mice, CD56^dim^ cells constituted the major NK cell subset (Figure 1), with a median (range) of 90.7% (89.6%–96.7%) among the total NK cells. However, most of CD56^dim^ NK cells were CD16^−^ (Figure 1), contrary to human reports [23], and consistent with a less mature state [24]. CD56^−^ and CD56^bright^ NK cells had a median (range) of 8.5% (3.3%–10%) and 0.2% (0%–1.7%) among total NK cells, respectively. As shown in Figure 3, compared with huNSG mice, huNOG-EXL mice had higher frequencies of circulating CD56^dim^ and CD56^−^ NK cells, with comparable frequencies of CD56^bright^ NK cells in blood between both mice strains. Notably, NK cell subsets were barely detectable in SLO (Figure 3). Particularly in the case of CD56^bright^ and CD56^−^ NK cells, we observed variability among huNOG-EXL mice, which can be related with their very low proportion among the total FSC-A^lo^ CD3^−^ cells. Also of note, NK cells might be maintained in huNOG-EXL mice via IL-15 production by DCs [25,26], which are efficiently engrafted in this mouse strain (see below), and could migrate to non-lymphoid tissues to exert immune surveillance [27].

Monocyte and DC subsets were also evaluated from FSC-A^hi^ CD3^−^ cells. In huNOG-EXL, the major monocyte population in blood was the CD14^+^ CD16^−^ (classical monocytes), with non-detectable frequencies of CD14^+^ CD16^+^ (intermediate monocytes) and CD14^−^ CD16^+^ cells (classical monocytes) (Figure 1 and Figure 4A, and data not shown). In addition, compared with huNSG mice, huNOG-EXL mice had higher frequencies of circulating CD14^+^ CD16^−^ monocytes (Figure 4A). However, the main localization of this subset was blood, with low to non-detectable frequencies in SLO (Figure 4A). These results partially agree with the reported frequencies of human monocytes in blood, where about 90% are classical monocytes and the CD16^+^ monocytes constitute the remaining cells [16], and with their blood residency [28]. The development of monocytes in huNOG-EXL is in accordance with the requirement of GM-CSF for monocytes/macrophages differentiation [29].

Dendritic cells’ development partially depends on GM-CSF [30]. Accordingly, huNOG-EXL exhibited higher frequencies of HLA-DR^+^ Lin 1^−^ cells (Figure 4B), where DCs are enriched. Among human DCs in blood, mDCs and pDCs constitute 60%–80% and 20%–40%, respectively [17,31]. However, consistent with the IL-3-dependent survival of pDCs [32,33], in huNOG-EXL (transgenic for human IL-3), pDCs constituted the major subset of circulating DCs, with a median (range) of 93.8% (84.4%–100%). Myeloid DCs comprised the 6.2% of total circulating DCs (range of 0%–15.6%). The lower development of mDCs could be related to their preferential dependence on the ligand for the fms-like tyrosine kinase (Flt3), in comparison with pDCs [34,35]. Indeed, Flt3 signaling is required for mDCs proliferation in the periphery [36]. Nonetheless, the frequencies of circulating and SLO-confined pDCs and mDCs were higher in huNOG-EXL compared with huNSG mice (Figure 4C and D). It is important to note that the predominance of pDCs in huNOG-EXL could impact the course of HIV infection or other diseases evaluated. For instance, pDCs are specialized in the release of type I interferon, which is critical for the anti-viral response and helps to polarize T-cell responses. However, pDCs are less efficient antigen-presenting cells than mDCs, which could hamper adaptive immune responses [37]. Thus, the huNOG-EXL mouse model recapitulates a pDC-predominant immune response, lacking full human DC features. 

In summary, compared with huNSG mice, huNOG-EXL exhibited higher levels of engraftment, as well as higher frequencies of lymphoid and myeloid populations, both in blood and SLO.

### 2.2. HIV Replication in huNOG-EXL Mice

Previous reports have demonstrated that huNOG mice support HIV replication [38,39,40]. However, to our knowledge, this is the first report evaluating HIV infection dynamics in huNOG-EXL mice. We determined the response of huNOG-EXL mice to an intraperitoneal HIV challenge with 15,000 Median Tissue Culture Infectious Dose (TCID_50_) units of the CCR5-tropic HIV reference BaL strain. As shown in Figure 5A, HIV-infected huNOG-EXL dramatically increased the viral load between the first and third weeks post-infection, with a stable viral load in the subsequent weeks of monitoring. As expected, no viral load was detected in uninfected mice. The increase in the viral load coincided with the decrease in the frequency of CD45^+^ cells in blood (Figure 5B), consistent with viral-induced suppression of the human hematopoietic progenitor cells [12]. In addition, there was an inversion of the CD4:CD8 ratio, reaching levels below 1 (Figure 5C), consistent with the decrease in the proportion of CD4^+^ T-cells due to a cytopathic viral effect. These changes were not observed in uninfected mice (Figure 5A–C). Importantly, the kinetics of HIV replication and decrease in the level of engraftment and CD4:CD8 ratio in huNOG-EXL mice were similar to those in HIV-infected huNSG mice (Figure 5D–F), and comparable to previous reports [12,13], although HIV-infected huNOG-EXL mice exhibited higher viral loads than huNSG mice in weeks 1–3 (*p* ≤ 0.009, data not shown). Thus, huNOG-EXL supports HIV replication.

### 2.3. Changes in Cell Populations along HIV Infection Time in huNOG-EXL Mice

We explored the changes that undergo lymphoid and myeloid cell populations in HIV-challenged huNOG-EXL mice along infection time, both in blood and SLO. We also compared the proportions of each cell subset between infected and uninfected huNOG-EXL mice at weeks 3, 5, and 7 post-infection, when the viral load had reached its peak and subsequently remained stable (Figure 5A). The frequencies of circulating T-cells and B-cells decreased along infection time in HIV-challenged mice (Figure 6A,B, left and middle panels), but not in SLO (Figure 6A,B, right panels). These results are consistent with the cytopathic viral effect [41], activation-induced cell death [42], and/or virus-induced cell migration to SLO [43,44,45,46]. Indeed, B-cells depletion in HIV is associated with increased susceptibility to CD95 ligand-mediated cell death [47], as well as intrinsic apoptosis [48]. CXCR5^+^ CD4^+^ T-cells, both circulating and follicle-confined, are relevant in HIV infection as they are a major target of this virus, constitute the main viral reservoir [49,50], and they have been found decreased in blood [51,52] but expanded in lymph nodes from HIV-infected patients [53]. However, there were no differences in the frequencies of CXCR5^+^ CD4^+^ T-cells between HIV-infected and uninfected huNOG-EXL mice, both in blood and SLO (Figure 6C). In contrast, similar to previous human reports [54], the frequencies of circulating CXCR5^+^ CD8^+^ T-cells were lower in HIV-infected huNOG-EXL mice compared with uninfected controls (Figure 6D, left and middle panels), but their frequency was higher in SLO (Figure 6D right panel), consistent with the recruitment of CXCR5^+^ CD8^+^ T-cells in SLO during HIV infection [55].

Myeloid subsets also suffered some changes in HIV-challenged huNOG-EXL mice. Similar to previous human reports [56], the frequencies of circulating CD14^+^ CD16^−^ classical monocytes decreased in infected huNOG-EXL mice compared with uninfected controls (Figure 7A, left and middle panels). Consistent with their low frequency in SLO (Figure 4A), there were no differences between infected and uninfected mice in the frequencies of classical monocytes in SLO (Figure 7A, right panel). Of note, the decrease in the proportion of classical monocytes in HIV-infected mice was not due to the increase of intermediate or non-classical monocytes, which remained at non-detectable proportions along infection time (data not shown). Probably, the decrease in classical monocytes is mediated by indirect mechanisms, since this subset is partially resistant to HIV infection, in comparison with macrophages [57,58]. Loss of CD14 upon monocyte activation may also account for the decrease in the proportion of these cells [59]. Nonetheless, a minor fraction of HIV-infected monocytes could constitute a latent viral reservoir [60].

Contrary to classical monocytes, the proportion of HLA-DR^+^ Lin 1^−^ cells in HIV-infected huNOG-EXL mice increased in blood along infection time (Figure 7B, left and middle panels), but decreased in SLO (Figure 7B, right panel), consistent with an egress of these cells to peripheral blood with infection in an attempt to restore the mature DC pool, or transitory migration to inflamed tissues [61]. Nonetheless, similar to previous human reports [62,63], we observed decreased frequencies of circulating pDCs and conserved frequencies of this subset in SLO (Figure 7C). There were no differences in the frequencies of mDCs between infected and uninfected huNOG-EXL mice (data not shown). The increase in HLA-DR^+^ Lin 1^−^ cells in HIV-infected huNOG-EXL mice occurred at expenses of CD11c^−^ CD123^−^ immature cells, since, in comparison with uninfected controls, they exhibited higher frequencies of this subset in blood (median [range] of 68.2 [45.5–87.5] vs. 42.8 [34.1–54.7], *p* = 0.01; data not shown), and lower frequencies in SLO (median [range] of 72.8 [38.4–90] vs. 83.4 [69.9–100], *p* = 0.04; data not shown). Thus, similar to previous human reports [64], HLA-DR^+^ Lin 1^−^ cells increase in blood from huNOG-EXL mice at expenses of the increase of CD11c^−^ CD123^−^ immature cells [65], whereas there is a decrease in the proportion of pDCs, which could be associated with viral infection and impaired function [66].

Lastly, we evaluated the changes in NK cell subsets. As shown in Figure 7D, contrary to previous human reports [67], there were no differences in the proportion of CD56^dim^ NK cells between HIV-infected and uninfected huNOG-EXL mice. However, there was a decrease in the proportion of circulating CD56^bright^ NK cells in HIV-infected huNOG-EXL mice (Figure 7E left and middle panels), but not in SLO (Figure 7E, right panel). Finally, there were no differences in the frequencies of CD56^−^ NK cells between HIV-infected and uninfected huNOG-EXL mice (data not shown). Considering the ability of CD56^bright^ NK cells to produce anti-viral cytokines, such as interferon-γ [15], their decrease could impair the activation of adaptive immune cells and enhance viral replication. Indeed, HIV infection induces phenotypic and functional changes of CD56^bright^ NK cells, which are associated with higher viral loads [68]. 

### 2.4. The Changes in Lymphoid and Myeloid Cells are Associated with the Level of HIV Replication in huNOG-EXL Mice

Together, the results presented here indicate that HIV infection induces changes in the proportions of circulating and SLO-confined lymphoid and myeloid populations. Of note, most of the changes occurred concomitantly with the increase in the viral load (from 1 to 3 weeks post-infection), and subsequently remained stable throughout the remaining monitoring time. This suggest that most of the alterations in human immune cell populations occur in an early phase of the infection and are maintained throughout the time while active HIV replication is present. Further, we analyzed the relationship between HIV replication and the changes in the cell subsets evaluated in blood and SLO. As shown in Table 1, the frequencies of circulating CD45^+^ cells, B-cells, CD14^+^ CD16^−^ classical monocytes, CD11c^−^ CD123^+^ pDCs, and CD56^bright^ NK cells, as well as the CD4:CD8 ratio, were negatively correlated with the plasma viral load. Interestingly, a positive correlation was found between the frequency of blood HLA-DR^+^ Lin 1^−^ cells and plasma viral load (Table 1), whereas there was a negative correlation when ALN-confined HLA-DR^+^ Lin 1^−^ cells were evaluated, consistent with a viral-induced increase in the proportion of this subset (mainly constituted by immature cells) in blood. In addition, the frequency of spleen CXCR5^+^ CD8^+^ T-cells positively correlated with plasma viral load (Table 1), suggesting that HIV replication, antigen levels, and/or a local inflammatory milieu induces the recruitment of this subset to SLO, as previously reported [55]. Finally, there was no correlation between the frequency of circulating total T-cells (Table 1), or other circulating or SLO-confined subsets (data not shown), and viral load. Altogether, these results support that the changes in several lymphoid and myeloid populations are associated with HIV replication, a product of active replication, bystander activation-induced cell death, or impaired cell maturation/differentiation. Cell redistribution in response to different antigenic burdens in lymphoid tissues may also drive the decrease in circulating populations. Certainly, huNOG-EXL mice gives relevant information regarding cellular dynamics during HIV infection.

## 3. Conclusions

Our results indicate that huNOG-EXL mice support HIV replication and recapitulate several viral-induced changes of human cell populations, such as the decrease in T-cells, B-cells, classical monocytes, and DCs. Thanks to the expression of human IL-3 and GM-CSF, huNOG-EXL mice exhibit enhanced and stable engraftment of myeloid populations. Thus, this model offers the possibility, in addition to the typical kinetics of viral replication and observable effects of anti-viral drugs, of studying other cell populations, which are not efficiently generated in other non-transgenic mouse strains, such as monocytes, macrophages, or DCs, that are relevant in the setting of HIV infection as viral reservoirs. The higher content of antigen-presenting cells in huNOG-EXL mice also allows myeloid-lymphoid cell interactions and potentially better adaptive immune responses. Nonetheless, some limitations were observed in the huNOG-EXL mouse model, such as the reconstitution of classical, but no other monocytes subsets, and a predominance of pDCs instead of mDCs. Thus, these issues should be considered for further studies evaluating HIV or other diseases.

## 4. Materials and Methods 

### 4.1. Ethics Statement

All animal care and procedures were performed according to protocols reviewed and approved by the Institutional Animal Care and Use Committee (IACUC) at the University of Maryland School of Medicine. Mice were monitored daily for morbidity and mortality, as previously reported [10].

### 4.2. Generation and HIV Infection of Humanized Mice

NOD.Cg-*Prkdc^scid^ Il2rg^tm1Sug^* Tg(SV40/HTLV-IL3,CSF2)10-7Jic/JicTac (NOG-EXL) mice were kindly provided by Taconic Biosciences (n = 6). NOD.Cg-*Prkcd^scid^ IL2rg^tmlWij^*/SzJ (NSG) mice were purchased from The Jackson Laboratory (n = 12). One to three-weeks old age female NOG-EXL and newborn NSG mice were gamma irradiated and engrafted with 1.2 × 10^5^ human umbilical cord blood-derived CD34^+^ hematopoietic stem cells (HSCs), obtained from a commercial provider, i.v. via the tail vein (NOG-EXL mice) or via intrahepatic injection (NSG mice). Mice were maintained with husbandry conditions and microbiological monitoring practices. Ten to fourteen weeks post-engraftment, mice were checked for human leukocyte reconstitution by flow cytometry. Mice with more than 25% of human CD45^+^ cells were intraperitoneally infected with 15,000 TCID_50_ units of the CCR5-tropic HIV reference BaL strain. Uninfected mice were included as controls. After infection, mice were consecutively euthanized every two weeks to obtain secondary lymphoid organs (SLO), by CO_2_ asphyxiation followed by cervical dislocation. 

### 4.3. Flow Cytometry

Peripheral blood (drawn from the retroorbital vein), spleen and lymph node mononuclear cells were obtained from mice (the latter when were available). Tissue samples were collected at necropsy and processed immediately in a 70 μm-pore size nylon cell strainer (Corning), followed by mononuclear cell isolation through a Ficoll density gradient (GE Healthcare). Remaining red blood cells were lysed with ACK buffer (Quality Biological Inc., Gaithersburg, MD, USA). For flow cytometry analysis, cells were incubated for 20 minutes at room temperature with optimized doses of the following anti-human antibodies: (i) T-cell/B-cell panel: CD3 APC (clone UCHT1, Biolegend), CD4 Alexa Fluor (AF) 488 (clone OKT4, Biolegend), CD8 AF 700 (clone OKT8, Thermo Fisher), CXCR5 PerCP Cy5.5 (clone RF8B2, BD), CD20 Brilliant Violet (BV) 605 (clone 2H7, Biolegend); (ii) Monocytes/NK cell panel: CD3 AF 700 (clone UCHT1, Biolegend), CD14 PE Cy7 (clone 63D3, Biolegend), CD16 APC-H7 (clone 3G8, BD Pharmingen), CD56 BV510 (clone 5.1H11, Biolegend); (iii) Dendritic cell panel: CD45 BV421 (clone 2D1, Biolegend), Lineage 1 cocktail FITC (CD3 [clone UCHT1], CD14 [clone HCD14], CD16 [clone 3G8], CD19 [clone HIB19], CD20 [clone 2H7] and CD56 [clone HCD56]; Biolegend), CD123 PE (clone 6H6, Biolegend), CD1c PE Cy7 (clone L161, Biolegend), HLA-DR PerCP Cy5.5 (clone LN3, Biolegend), CD11c BV605 (clone 3.9, Biolegend). After incubation, red cells from peripheral blood were lysed with BD FACS Lysing Solution (BD), and all samples were washed twice with FACS buffer (1% Fetal Bovine Serum in 1X PBS). Finally, cells were fixed with 1% paraformaldehyde in 1X PBS, and acquired on a FACSAria II cytometer (BD) within an hour of completing the staining. Data were analyzed with the FlowJo Software version 8.7 (Tree Star, Inc., Ashland, OR, USA).

### 4.4. Determination of Viral Load

Plasma was obtained for quantification of HIV RNA copy number by an in-house real-time RT-PCR, using HIV gag primers, SK38/SK39 and SYBR green dyes, as previously reported [10,11]. The assay has a sensitivity of 150 HIV RNA copies/40 μL plasma.

### 4.5. Statistical Analysis

GraphPad Prism software v. 7.0 (GraphPad Software, La Jolla, CA, USA) was used for statistical analysis. Data are presented as medians and ranges. The Mann–Whitney test was used for comparison of 2 independent data. The degree of correlation between variables was determined with the Spearman test. In all the analyses, a value equal to the half of the limit of detection of the assay was assigned to samples with non-detectable viral load. In all cases, a *p* value < 0.05 was considered significant.

## Figures and Tables

**Figure 1 pathogens-08-00033-f001:**
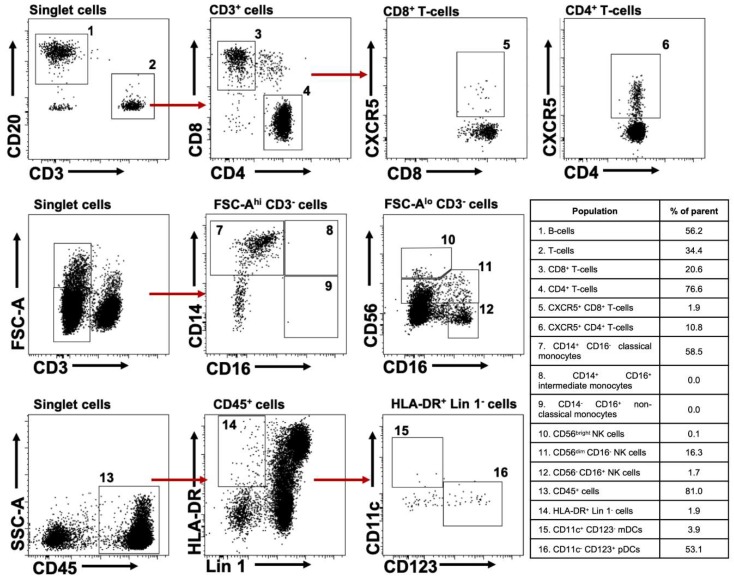
Cell subsets evaluated in huNOG-EXL mice. Representative gating strategy from blood cells for the identification of the cell populations evaluated. The number next to the gates represents the respective cell subset found in the adjacent table.

**Figure 2 pathogens-08-00033-f002:**
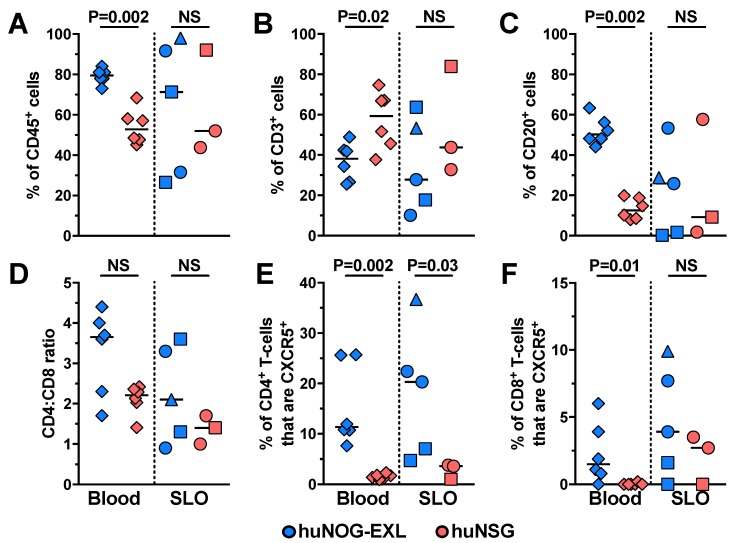
huNOG-EXL mice exhibit an efficient engraftment of lymphoid populations. Frequencies of CD45^+^ (**A**), CD3^+^ (**B**), and CD20^+^ (**C**) cells from total singlet cells, the CD4:CD8 ratio (**D**), and the frequencies of CD4^+^ (**E**) and CD8^+^ cells (**F**) that are CXCR5^+^ in blood (diamonds) and secondary lymphoid organs (SLO; spleen: circles; axillary lymph node: squares; mesenteric lymph node: triangles) from huNOG-EXL and huNSG mice. The *p* value of the Mann-Whitney test is shown. NS: Not statistically significant.

**Figure 3 pathogens-08-00033-f003:**
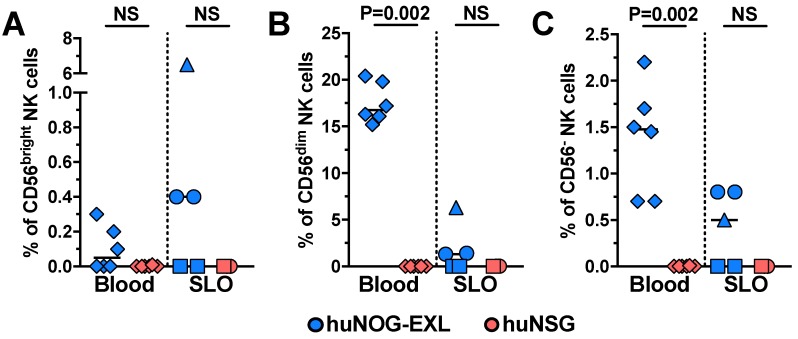
huNOG-EXL mice have higher levels of NK cells than huNSG mice. Frequencies of CD56^bright^ (**A**), CD56^dim^ (**B**), and CD56^−^ (**C**) NK cells (from FSC-A^lo^ CD3^−^ cells) in blood (diamonds) and secondary lymphoid organs (SLO; spleen: circles; axillary lymph node: squares; mesenteric lymph node: triangles) from huNOG-EXL and huNSG mice. The *p* value of the Mann-Whitney test is shown. NS: Not statistically significant.

**Figure 4 pathogens-08-00033-f004:**
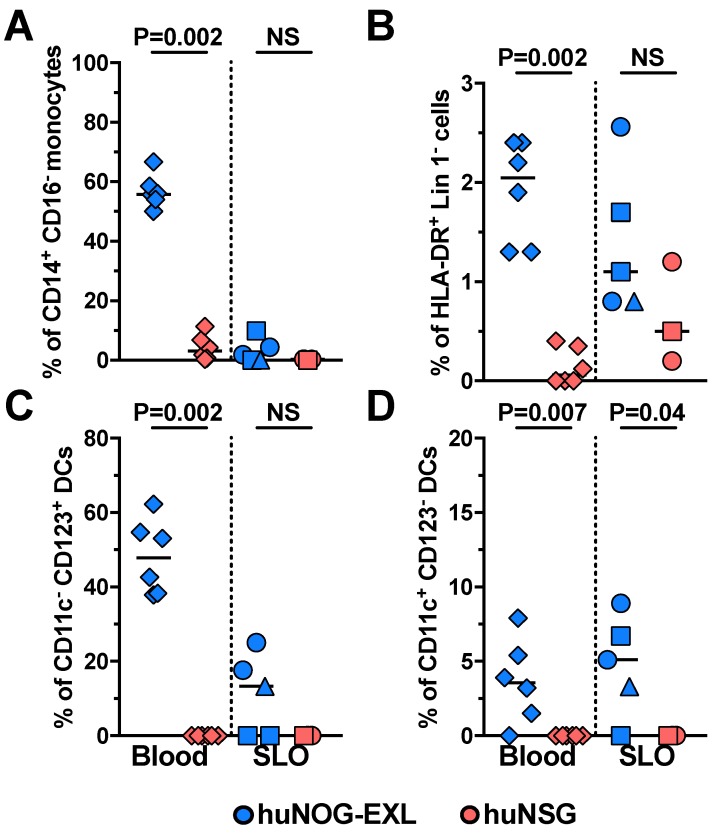
huNOG-EXL mice exhibit an efficient engraftment of myeloid populations. Frequencies of CD14^+^ CD16^−^ (classical) monocytes from FSC-A^hi^ CD3^−^ cells (**A**), HLA-DR^+^ Lin 1^−^ cells from CD45^+^ cells (**B**), CD11c^−^ CD123^+^ plasmacytoid dendritic cells (**C**), and CD11c^+^ CD123^−^ myeloid dendritic cells (**D**), the latter from HLA-DR^+^ Lin 1^−^ cells, in blood (diamonds) and secondary lymphoid organs (SLO; spleen: circles; axillary lymph node: squares; mesenteric lymph node: triangles) from huNOG-EXL and huNSG mice. The *p* value of the Mann-Whitney test is shown. NS: Not statistically significant.

**Figure 5 pathogens-08-00033-f005:**
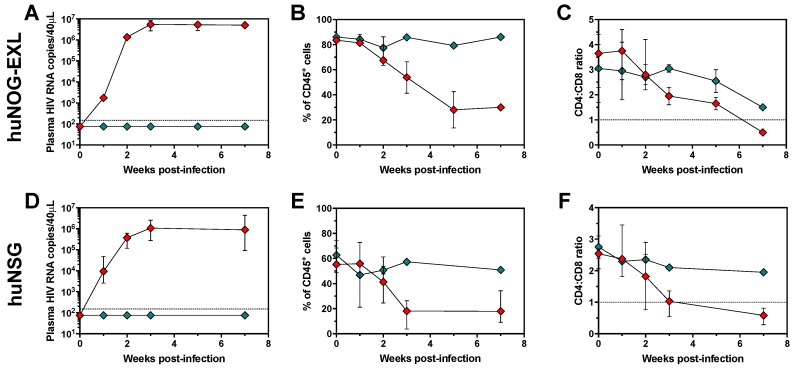
huNOG-EXL mice support the replication of HIV, with the consequent decrease in the level of engraftment and inversion of the CD4:CD8 ratio. Viral load (**A**,**D**), blood frequencies of CD45^+^ cells (**B**,**E**), and blood CD4:CD8 ratio (**C**,**F**) in huNOG-EXL (**A**–**C**) and huNSG (**D**–**F**) mice after infection with HIV (green diamonds: uninfected mice; red diamonds: HIV-infected mice). In **A** and **D**, the dashed lines indicate the limit of detection of the assay. In **C** and **F**, the dashed lines indicate the CD4:CD8 ratio = 1.

**Figure 6 pathogens-08-00033-f006:**
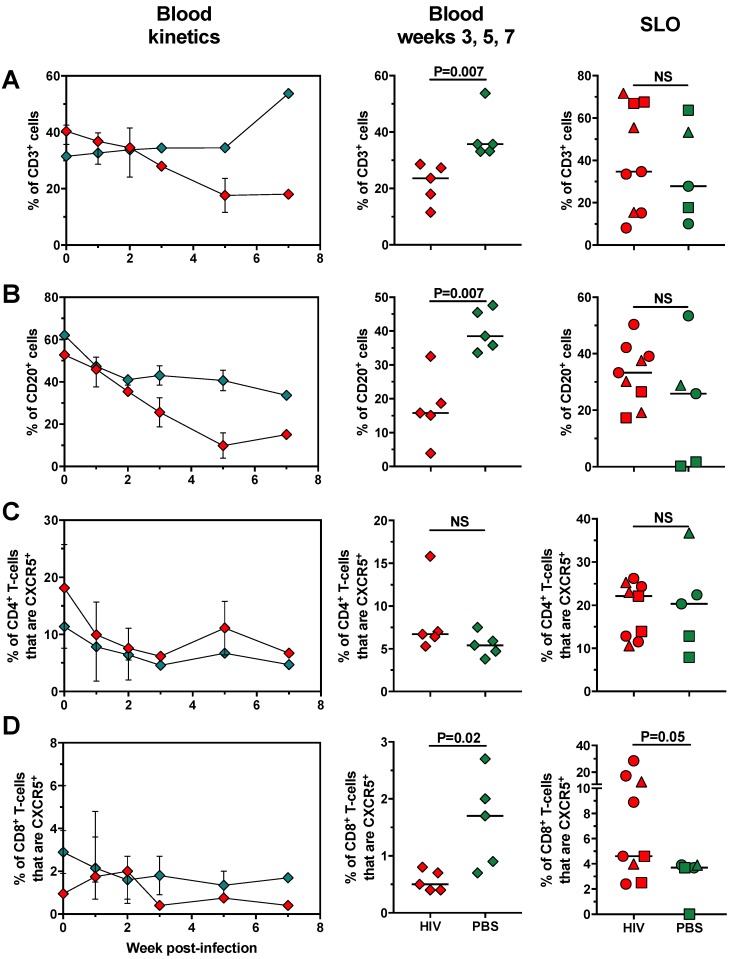
HIV infection in huNOG-EXL mice induces the decrease of circulating T-cells, B-cells, and CXCR5^+^ CD8^+^ T-cells. Frequencies of CD3^+^ cells (**A**) and CD20^+^ cells (**B**) from total singlet cells, CXCR5^+^ CD4^+^ T-cells (**C**), and CXCR5^+^ CD8^+^ T-cells (**D**) from total CD4^+^ and CD8^+^ T-cells, respectively, in huNOG-EXL mice after infection with HIV. In the left panels, the kinetics of blood populations are shown; in the middle panels, comparisons of blood populations between infected and uninfected mice in compiled data from 3, 5, and 7 weeks post-infection are shown; in the right panels, comparisons of secondary lymphoid organs (SLO; spleen: circles; axillary lymph node: squares; mesenteric lymph node: triangles)-confined populations between infected and uninfected mice are shown. In all the cases, green: uninfected mice; red: HIV-infected mice. The *p* value of the Mann-Whitney test is shown.

**Figure 7 pathogens-08-00033-f007:**
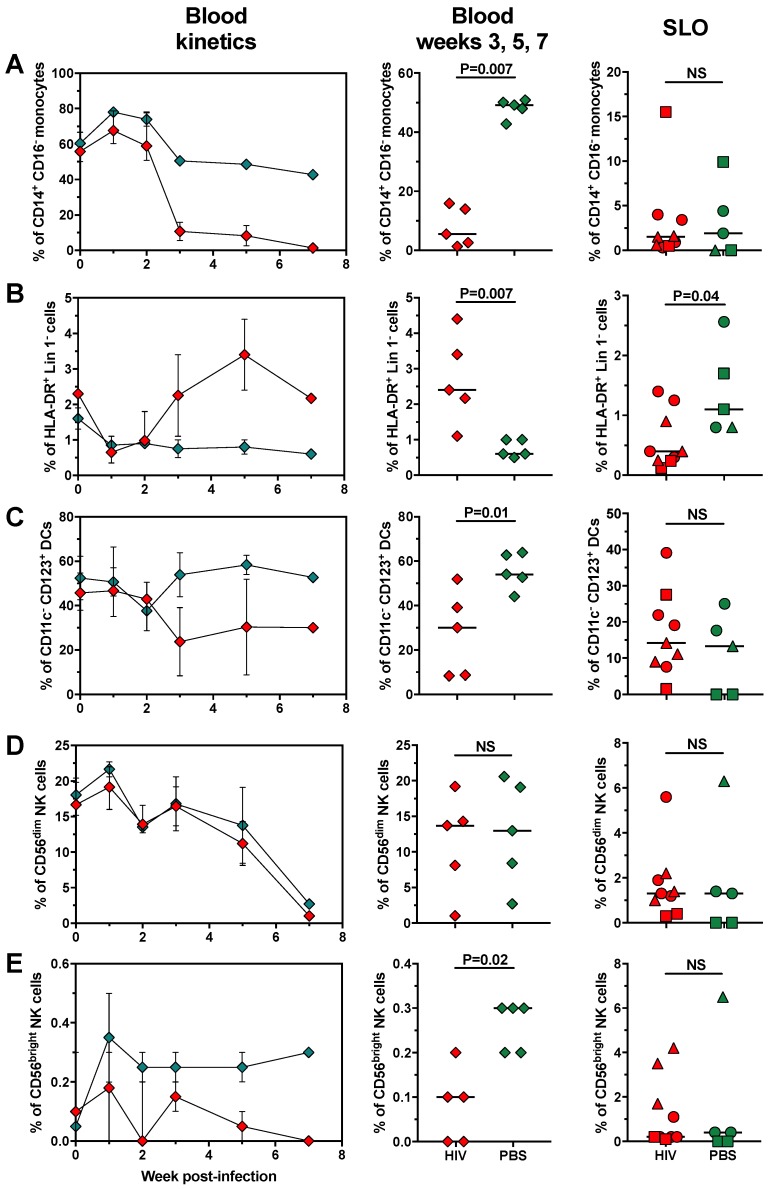
HIV infection in huNOG-EXL mice affects the frequencies of classical monocytes, dendritic cells, and CD56^bright^ NK cells. Frequencies of CD14^+^ CD16^−^ (classical) monocytes from FSC-A^hi^ CD3^−^ cells (**A**), HLA-DR^+^ Lin 1^−^ cells from CD45^+^ cells (**B**), CD11c^−^ CD123^+^ plasmacytoid dendritic cells from HLA-DR^+^ Lin 1^−^ cells (**C**), CD56^dim^ (**D**), and CD56^bright^ NK cells (**E**), the latter from FSC-A^lo^ CD3^−^ cells, in huNOG-EXL mice after infection with HIV. In the left panels, the kinetics of blood populations are shown; in the middle panels, comparisons of blood populations between infected and uninfected mice in compiled data from 3, 5, and 7 weeks post-infection are shown; in the right panels, comparisons of secondary lymphoid organs (SLO; spleen: circles; axillary lymph node: squares; mesenteric lymph node: triangles)-confined populations between infected and uninfected mice are shown. In all the cases, green: uninfected mice; red: HIV-infected mice. The *p* value of the Mann-Whitney test is shown.

**Table 1 pathogens-08-00033-t001:** Correlations between viral load and cell populations in blood and SLO.

Correlations Plasma Viral Load vs. Cell Populations		Spearman Test	
Compartment ^1^	Population	rho	*p* Value
Blood	CD45^+^ cells	−0.89	0.0005
Blood	CD3^+^ T-cells	−0.3	0.1
Blood	CD20^+^ B-cells	−0.85	0.02
Blood	CD4:CD8 ratio	−0.66	0.02
Spleen	CXCR5^+^ CD8^+^ T-cells	0.92	0.02
Blood	HLA-DR^+^ Lin 1^−^ cells	0.91	0.0004
Axillary lymph node	HLA-DR^+^ Lin 1^−^ cells	−0.73	0.07
Blood	CD14^+^ CD16^−^ classical monocytes	−0.88	0.001
Blood	CD11c^−^ CD123^+^ pDCs	−0.87	0.002
Blood	CD56^bright^ NK cells	−0.72	0.02

^1^ In blood, weeks 3, 5, and 7 post-infection are analyzed.

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
