# Peer review of "HIV Replication in Humanized IL-3/GM-CSF-Transgenic NOG Mice"

_pathogens, 2019, doi:10.3390/pathogens8010033_

Round 1

Reviewer 1 Report

1. This research study has serious flaws in the experimental design.  First, the authors have not carefully controlled their studies; there are too many variable involved to make solid conclusions.  The first such error is in comparing huNOG-EXL mice to huNSG mice.  Although the differences in NOG and NSG mice are likely subtle, the study should have been performed with NOG vs NOG-EXL mice.  In addition, the Methods indicate that NOG-EXL mice were engrafted via tail vein injection and NSG mice were engrafted via intrahepatic injection of human cells.  Thus, there are now 3 variables that exist between the 2 main study groups:  mouse background strain, inclusion of human knock-in genes, and route of human cell engraftment.  Figures 2, 3 and 4 should be re-done with the only variable being the inclusion/exclusion of the human knock-in genes.  The authors should also comment about the origin of the human CD34+ cells.  If different donors were used for the different mouse strains, that can also affect the results because not all donor cells engraft with equal efficiency into immunodeficient mice, depending upon the donor genotype. 

2. There is no indication of the origin of the cells for Figure 1.  Do the cells come from blood? 

3. In Figure 3, there are panels that show blood or "Secondary Lymphoid Organs", which turns out to be a combination of spleen, axillary lymph node, and mesenteric lymph node.  However, there are only mesenteric lymph node results shown for huNOG-EXL mice and not for huNSG mice.  I don't believe the results of this study because it isn't appropriate to just lump together all of the secondary lymphoid organs, especially when the 3 different types are not represented amongst the two types of animals.  The single MLN sample in Fig 2E appears to skew the results to a significant p value, but if a single MLN was present in the control then it might be as equally high. 

4.  The gating strategy for the flow cytometry plots needs to be explained further.  This occurs multiple times in the paper figures and needs to be fixed, because the reader cannot understand what cell populations are being analyzed without being presented with the gating strategy.  One prominent example is Fig 7, where it isn't clear to me what the y axis is showing in Fig 7A.  It shows a "% of CD14+CD16- monocytes", and the results show a range beginning at 50-60% and then decreasing at later time points.  Are these data showing the total monocyte population?  If so, what stains were used to determine that these are monocytes, so that the reader can determine if the results accurately depict an actual monocyte population?  It clearly isn't a CD45+ population because 50-60% is far too high, but no explanation is given. 

5.  The authors should comment on why some animals produce variable levels of certain cell types as compared to other animals in the same group that do not produce any (e.g., Fig. 3 shows high variability in NK cells in huNOG-EXL mice). 

6.  At the beginning of section 2.2, the authors should mention the HIV strain used, the titer of virus inoculated, and the method of infection.  All of these variables affect the ensuing pathogenesis and this information should be provided in the Results. 

7.  On line 137, it appears that the reference to Fig 5C should be corrected to Fig 5B. 

8.  The authors state on line 140 that no inversion of the CD4:CD8 ratio was observed in uninfected animals.  However, the pattern seen in the results depicted in Fig 5C show a similar rate of decrease in both groups at later time points (weeks 3-7). 

9.  Throughout the paper, the authors should clarify when human-specific antibodies are used in FACS staining.  For example, hCD45+ cells instead of just CD45+ cells.  These mice have many of these same cell types and markers, and this should be specified. 

10.  Was there a significant difference in the viral load between huNOG-EXL mice and huNSG mice?  There appears to be nearly a 10-fold increase. 

11.  It seems odd to combine the results for weeks 3/5/7 for blood samples, especially since the values are changing in the first panels.  What was the rationale for reporting the results this way?  It seems like an attempt to achieve statistical significance for the results when the sample size was small. 

12.  In the SLO panel of Fig 6D, it appears that some organs had a decrease in CD8+CXCR5+ cells, but others did not, relative to the control.  The authors show a p value of 0.05, but that doesn't seem to tell the whole story.  Why would some change, and others not? 

Author Response

We would like to acknowledge this reviewer for the time invested in reading this manuscript. The following are our answers to each comment.

1. This research study has serious flaws in the experimental design.  First, the authors have not carefully controlled their studies; there are too many variable involved to make solid conclusions.  The first such error is in comparing huNOG-EXL mice to huNSG mice.  Although the differences in NOG and NSG mice are likely subtle, the study should have been performed with NOG vs NOG-EXL mice.  In addition, the Methods indicate that NOG-EXL mice were engrafted via tail vein injection and NSG mice were engrafted via intrahepatic injection of human cells.  Thus, there are now 3 variables that exist between the 2 main study groups:  mouse background strain, inclusion of human knock-in genes, and route of human cell engraftment.  Figures 2, 3 and 4 should be re-done with the only variable being the inclusion/exclusion of the human knock-in genes.  The authors should also comment about the origin of the human CD34+ cells.  If different donors were used for the different mouse strains, that can also affect the results because not all donor cells engraft with equal efficiency into immunodeficient mice, depending upon the donor genotype.  

R/         We agree with the reviewer on the subtle difference between NOG and NSG mice. These strains harbor two mutations that make them immunodeficient. Both NOG and NSG share the Prkdcscid mutation but differ in the IL2rg mutation. While NOG strain has a mutation in IL-2rg gene that is translated in the production of an IL2rg lacking the intracytoplasmic domain making it nonfunctional, NSG mice are knockout for IL2rg, meaning that this protein is not produced at all (see discussion in line 56-59). Therefore, the immuno-deficient phenotype of both strains is similar, allowing the reconstitution of human CD45+ cells after CD34 engraftment at levels higher than 50% (Figure 2A). Regarding the route of inoculation, we used our in house-standardize NSG model as a reference (intrahepatic humanized) to analyze the commercially available NOG-EXL mice from TACONIC (intravenously humanized) (see added discussion in line 59-60). We are certain that the development of myeloid lineage is due to the expression IL-3 and GM-CSF human genes in NOG-EXL mice as reported by TACONIC and (Fukuchi et al, 1998, Ito R et al 2013) not due to the use of different route for cell engraftment. Finally, we included humanized-NSG mice as a reference, because this model is been extensively used by us and other researchers for HIV studies, particularly as a chronic model of infection (see added discussion in line 60-61). However, if we take out the results obtained from NSG mice from this manuscript, it does not modify the results or conclusions obtained from NOG-EXL mice, showing that these animals reconstituted human immune cells after hCD34 engraftment, they developed and constantly maintained myeloid lineage, and they supported HIV chronic infection making them a suitable model to study the role of myeloid cells in the context of HIV infection. CD34+ cells are from two different donors. One for NSG and other for NOG-EXL. However, all animals were screened 12 weeks after engraftment to assure acceptable levels of human cells. In this study, those animals showing poor levels engraftment were excluded. 

2. There is no indication of the origin of the cells for Figure 1.  Do the cells come from blood? 

R/         Yes. They do. The gating strategy shows blood cells (see Fig 1 legend).

3. In Figure 3, there are panels that show blood or "Secondary Lymphoid Organs", which turns out to be a combination of spleen, axillary lymph node, and mesenteric lymph node.  However, there are only mesenteric lymph node results shown for huNOG-EXL mice and not for huNSG mice.  I don't believe the results of this study because it isn't appropriate to just lump together all of the secondary lymphoid organs, especially when the 3 different types are not represented amongst the two types of animals.  The single MLN sample in Fig 2E appears to skew the results to a significant p value, but if a single MLN was present in the control then it might be as equally high.

R/         Considering that not all the animals develop axillary or mesenteric lymph nodes, and to perform statistical comparisons, we pooled the data of spleen, axillary and mesenteric lymph nodes, the latter when available. In addition, since our number of mice included is <30 and the data do not follow a normal distribution, we show median and ranges and performed non-parametric analyses. For instance, if Fig 2E, when we remove the value of mesenteric lymph node in huNOG-EXL, the P=0.05 (Mann-Whitney test), indicating that a tendency for a higher frequency of CXCR5+ CD4+ T-cells is present in huNOG-EXL compared with huNSG mice.

4.  The gating strategy for the flow cytometry plots needs to be explained further.  This occurs multiple times in the paper figures and needs to be fixed because the reader cannot understand what cell populations are being analyzed without being presented with the gating strategy.  One prominent example is Fig 7, where it isn't clear to me what the y-axis is showing in Fig 7A.  It shows a "% of CD14+CD16- monocytes", and the results show a range beginning at 50-60% and then decreasing at later time points.  Are these data showing the total monocyte population?  If so, what stains were used to determine that these are monocytes, so that the reader can determine if the results accurately depict an actual monocyte population?  It clearly isn't a CD45+ population because 50-60% is far too high, but no explanation is given.

R/         As suggested by the reviewer, we have clarified the gating used for the presented data in all the figures. See Fig 2, 3, 4, 6 and 7 legends.

5.  The authors should comment on why some animals produce variable levels of certain cell types as compared to other animals in the same group that do not produce any (e.g., Fig. 3 shows high variability in NK cells in huNOG-EXL mice).

R/         As suggested by the reviewer, we have included a discussion on the variability of NK cell frequencies (see line 97-99).

6.  At the beginning of section 2.2, the authors should mention the HIV strain used, the titer of virus inoculated, and the method of infection.  All of these variables affect the ensuing pathogenesis and this information should be provided in the Results.

R/         As suggested by the reviewer, we added the description of the HIV challenge in lines 151-152.

7.  On line 137, it appears that the reference to Fig 5C should be corrected to Fig 5B.

R/         In agreement with the reviewer, we changed Fig 5C for 5B in line 156.

8.  The authors state on line 140 that no inversion of the CD4:CD8 ratio was observed in uninfected animals.  However, the pattern seen in the results depicted in Fig 5C show a similar rate of decrease in both groups at later time points (weeks 3-7).

R/         In agreement with the reviewer, we show a decrease in the CD4:CD8 ratio in control mice, but they did not reach levels below 1 (which is a typical finding during HIV infection) compared with HIV-infected mice.

9.  Throughout the paper, the authors should clarify when human-specific antibodies are used in FACS staining.  For example, hCD45+ cells instead of just CD45+ cells.  These mice have many of these same cell types and markers, and this should be specified.

R/         As suggested by the reviewer, we clarified that human populations were evaluated with the use of anti-human monoclonal antibodies; see line 53-54.

10.  Was there a significant difference in the viral load between huNOG-EXL mice and huNSG mice?  There appears to be nearly a 10-fold increase.  

R/         Based on the reviewer comment we compared the viral load between HIV-infected huNOG-EXL and huNSG mice, and huNOG-EXL exhibited higher viral load in weeks 1-3 (see lines 162-163).

11.  It seems odd to combine the results for weeks 3/5/7 for blood samples, especially since the values are changing in the first panels.  What was the rationale for reporting the results this way?  It seems like an attempt to achieve statistical significance for the results when the sample size was small.

R/         As described in lines 175-177, we compared the data of weeks 3-7 since the viral load had reached its peak and subsequently remained stable, allowing us to determine the changes in the populations and possible effects of viral replication, compared with uninfected mice.

12.  In the SLO panel of Fig 6D, it appears that some organs had a decrease in CD8+CXCR5+ cells, but others did not, relative to the control.  The authors show a p value of 0.05, but that doesn't seem to tell the whole story.  Why would some change, and others not?  

R/         As described in lines 267-269, we observed a positive correlation between plasma viral load and the frequency of spleen CXCR5+CD8+T-cells, so that the viral replication and/or local inflammation may be driving the redistribution of this subset. Thus, in Fig 6D, the high values in spleen are those mice with high viral load.

Reviewer 2 Report

The manuscript by Perdomo-Celis et al. describes engraftment and HIV infection of humanized NOG-EXL mice.  The manuscript contains much detail about individual human cell types in these mice and is well-written.  Although there are multiple humanized mouse models for HIV infection, this study still adds an important aspect to the field. The manuscript could be improved by changing the presentation of the data, adding additional statistical analyses, and expanding the discussion.

Major points

1)  Throughout the manuscript and in all the figures, it is confusing what "percentage" means when referring to human immune cell subsets.  Some simple clarification in the figures or in the figure legend is necessary (for example, in Figure 2B, is "% of CD3+ cells" the percentage of CD3+ cells in the human CD45+ gate?).

2)  In Figures 2-4, combining the spleens and lymph nodes in a single category is problematic, as there may be differences between these tissues.  It is recommended that they be separated in the figures.

3)  In Figures 5-7, please change the graph data points so it is easier to visualize between infected and uninfected samples.  Also, it is important to add statistical analysis to all parts of these figures.

4)  The conclusion section needs to include in-depth discussion of how the NOG-EXL HIV model compares to the multiple other humanized mouse models for HIV infection.

Minor points

1)  Please briefly discuss the differences between NOG and NSG mice.

2)  Which tissue is Figure 1 showing?

3)  Lines 166 and 181, please clarify that these references are to human studies, not previously published NOD-EXL studies.

Author Response

We would like to acknowledge this reviewer for the time invested in reading this manuscript. The following are our answers to each comment.

1)  Throughout the manuscript and in all the figures, it is confusing what "percentage" means when referring to human immune cell subsets.  Some simple clarification in the figures or in the figure legend is necessary (for example, in Figure 2B, is "% of CD3+ cells" the percentage of CD3+ cells in the human CD45+ gate?).

R/         As suggested by the reviewer, we have clarified the gating used for the presented data in all the figures. See Fig 2, 3, 4, 6 and 7 legends.

2)  In Figures 2-4, combining the spleens and lymph nodes in a single category is problematic, as there may be differences between these tissues.  It is recommended that they be separated in the figures.

R/         We agree with this reviewer and were aware of the differences between lymphoid tissues. However, considering that not all the animals develop axillary or mesenteric lymph nodes, and to perform statistical comparisons, we pooled the data of spleen, axillary and mesenteric lymph nodes, the latter when available. Indeed, only one axillary lymph node was available in huNSG mice, so that, if they are separated, it would make more difficult to interpret the results.

3)  In Figures 5-7, please change the graph data points so it is easier to visualize between infected and uninfected samples.  Also, it is important to add statistical analysis to all parts of these figures.

R/         As suggested by the reviewer, we have enlarged the dots in all the figures for easier visualization. Regarding the statistical analysis, it should be noted that only 2 mice were left uninfected, restraining a statistical comparison. Thus, we pooled data of weeks 3-7, when the viral load had reached its peak and subsequently remained stable, allowing us to determine the changes in the populations and possible effects of viral replication, compared with uninfected mice.

4)  The conclusion section needs to include in-depth discussion of how the NOG-EXL HIV model compares to the multiple other humanized mouse models for HIV infection.

R/         As suggested by the reviewer, we extended the discussion of how the NOG-EXL HIV model compares to other humanized mouse models (line 282-291).

Minor points

1)  Please briefly discuss the differences between NOG and NSG mice.

R/         Ass suggested, we added this discussion in lines 56-59.

2)  Which tissue is Figure 1 showing?

R/         The gating strategy shows blood cells (see Fig 1 legend).

3)  Lines 166 and 181, please clarify that these references are to human studies, not previously published NOD-EXL studies.

R/         Ass suggested, we clarified that these are human studies (line 187 and 203)

Reviewer 3 Report

Perdomo-Celis et. al., provide data supporting the development of a new humanized mouse model for HIV infection.  The model appears to recapitulate many features not well represented by other humanized mouse models of HIV and thus may have significant benefit for understanding the pathogenesis of HIV as well as conceivably contribute to the development of medical countermeasures including vaccines or therapeutics.  The work is clearly written and data well presented. 

A few minor points that may help to improve the interpretation of the work are suggested below:

Line 90--Should be "Also of note," or similiar

Lines 82-91--Any potential for NK populations to be distributed differentially in other non-lymphoid compartments?  New residencies?

Line 104--How do these frequencies compare with humans?

Lines 115-125--Please elaborate on how or if you think the pDC vs mDC populations in these mice, which are not equivalently populated to humans, impact the course of infection in this model.  Pure description here is useful, but makes interpretation challenging as to why you included it or even looked at these populations.

Line 139--Change to "below 1".

Histology might have been a powerful piece of data to assist in enumerating the different cell populations you were looking for.  Not mandatory for publication, but the authors should consider this in the future.

Lines 218-223--Do the authors believe this lack of NK related changes may have influenced the results they observed with respect ot the course of infection and differential recruitment of immune populations, rate of viral clearance (or lack thereof), or even survival?

Line 227--"our previous"  syntax issue, do you mean the previous work from your lab...or all the data discussed in this work?  Consider rewording.

Conclusions--This section is a little thin.  At minimum, also discuss some of the shortcomings of this model and whether these shortcomings would impact medical countermeasure development.

Author Response

We would like to acknowledge this reviewer for the time invested in reading this manuscript. The following are our answers to each comment.

A few minor points that may help to improve the interpretation of the work are suggested below:

Line 90--Should be "Also of note," or similar

R/         As suggested by the reviewer, we changed this phrase for “Also of note” (line 99).

Lines 82-91--Any potential for NK populations to be distributed differentially in other non-lymphoid compartments?  New residencies?

R/         In agreement with the reviewer, NK cells could potentially migrate to other non-lymphoid tissues for immune surveillance. We added this discussion in lines 100-101.

Line 104--How do these frequencies compare with humans?

R/         As suggested by the reviewer, we have added a discussion comparing our monocyte results with human reports (lines 115-117)

Lines 115-125--Please elaborate on how or if you think the pDC vs mDC populations in these mice, which are not equivalently populated to humans, impact the course of infection in this model.  Pure description here is useful, but makes interpretation challenging as to why you included it or even looked at these populations.

R/         As suggested by the reviewer, we have added a discussion regarding the predominance of pDCs in huNOG-EXL mice (lines 138-143).

Line 139--Change to "below 1".

R/         As suggested, we changed it to “below 1” (line 158).

Histology might have been a powerful piece of data to assist in enumerating the different cell populations you were looking for.  Not mandatory for publication, but the authors should consider this in the future.

R/         Here, we were not able to perform histological analyses. However, in agreement with the reviewer, we will consider them for future studies. 

Lines 218-223--Do the authors believe this lack of NK related changes may have influenced the results they observed with respect to the course of infection and differential recruitment of immune populations, rate of viral clearance (or lack thereof), or even survival?

R/         As suggested by the reviewer, we have added a discussion regarding the HIV-induced changes in NK cells (lines 246-250).

Line 227--"our previous" syntax issue, do you mean the previous work from your lab...or all the data discussed in this work?  Consider rewording.

R/         As suggested, we changed “our previous” for “the results here presented” (line 254).

Conclusions--This section is a little thin.  At minimum, also discuss some of the shortcomings of this model and whether these shortcomings would impact medical countermeasure development.

R/         As suggested, we have extended the conclusions section, highlighting the advantages and limitations of the huNOG-EXL model (line 282-291).

Round 2

Reviewer 1 Report

The data are presented much more clearly in this version. 

Reviewer 2 Report

The authors have addressed most of this reviewer's concerns.